# Grape Pomace Extract Attenuates Inflammatory Response in Intestinal Epithelial and Endothelial Cells: Potential Health-Promoting Properties in Bowel Inflammation

**DOI:** 10.3390/nu14061175

**Published:** 2022-03-11

**Authors:** Nadia Calabriso, Marika Massaro, Egeria Scoditti, Tiziano Verri, Amilcare Barca, Carmela Gerardi, Giovanna Giovinazzo, Maria Annunziata Carluccio

**Affiliations:** 1National Research Council (CNR) Institute of Clinical Physiology (IFC), 73100 Lecce, Italy; marika.massaro@ifc.cnr.it (M.M.); egeria.scoditti@ifc.cnr.it (E.S.); 2Department of Biological and Environmental Sciences and Technologies (DISTEBA), University of Salento, 73100 Lecce, Italy; tiziano.verri@unisalento.it (T.V.); amilcare.barca@unisalento.it (A.B.); 3National Research Council (CNR) Institute of Sciences of Food Production (ISPA), 73100 Lecce, Italy; carmela.gerardi@ispa.cnr.it (C.G.); giovanna.giovinazzo@ispa.cnr.it (G.G.)

**Keywords:** gut inflammation, endothelial dysfunction, pro-inflammatory markers, leukocyte adhesion, grape pomace, polyphenols, oxidative stress, gene expression

## Abstract

Inflammatory bowel disease (IBD) implies the chronic inflammation of the gastrointestinal tract, combined with systemic vascular manifestations. In IBD, the incidence of cardiovascular disease appears to be related to an increase of oxidative stress and endothelial dysfunction. Grape pomace contains high levels of anti-oxidant polyphenols that are able to counteract chronic inflammatory symptoms. The aim of this study was to determine whether grape pomace polyphenolic extract (GPE) was able to mitigate the overwhelming inflammatory response in enterocyte-like cells and to improve vascular function. Intestinal epithelial Caco-2 cells, grown in monolayers or in co-culture with endothelial cells (Caco-2/HMEC-1), were treated with different concentrations of GPE (1, 5, 10 µg/mL gallic acid equivalents) for 2 h and then stimulated with lipopolysaccharide (LPS) and tumor necrosis factor (TNF)-α for 16 h. Through multiple assays, the expression of intestinal and endothelial inflammatory mediators, intracellular reactive oxygen species (ROS) levels and NF-κB activation, as well as endothelial-leukocyte adhesion, were evaluated. The results showed that GPE supplementation prevented, in a concentration-dependent manner, the intestinal expression and release of interleukin (IL)-6, monocyte chemoattractant protein (MCP)-1, and matrix metalloproteinases (MMP)-9 and MMP-2. In Caco-2 cells, GPE also suppressed the gene expression of several pro-inflammatory markers, such as IL-1β, TNF-α, macrophage colony-stimulating factor (M-CSF), C-X-C motif ligand (CXCL)-10, intercellular adhesion molecule (ICAM)-1, vascular cell adhesion molecule (VCAM)-1, and cyclooxygenase (COX)-2. The GPE anti-inflammatory effect was mediated by the inhibition of NF-κB activity and reduced intracellular ROS levels. Furthermore, transepithelial GPE suppressed the endothelial expression of IL-6, MCP-1, VCAM-1, and ICAM-1 and the subsequent adhesion of leukocytes to the endothelial cells under pro-inflammatory conditions. In conclusion, our findings suggest grape pomace as a natural source of polyphenols with multiple health-promoting properties that could contribute to the mitigation of gut chronic inflammatory diseases and improve vascular endothelial function.

## 1. Introduction

Cardiovascular diseases (CVD) represent the main cause of mortality and morbidity worldwide, due to the continuous increase in the prevalence of cardiovascular risk factors. Several studies have shown heightened risk of cardiovascular complications in chronic inflammatory disorders, especially those affecting the gastrointestinal tract, such as inflammatory bowel diseases (IBD) [1,2,3]. IBD and CVD share similar immune response in chronic systemic inflammation and atherogenesis [4]. In the course of IBD, increasing concentrations of pro-inflammatory cytokines may lead to endothelial dysfunction and CVD development [5]. Endothelial dysfunction is a systemic disorder characterized by imbalanced vasodilation and vasoconstriction, elevated reactive oxygen species (ROS), and pro-inflammatory factors, as well as deficiency of nitric oxide bioavailability [6,7]. In particular, in both CVD and IBD, pro-inflammatory angiogenesis is recognized as a common trait, sustaining both atherosclerotic plaque growth and intestinal inflammation [8,9]. The inflamed intestinal mucosa itself seems to play an important role in promoting arterial disease [10]. IBD patients have a disrupted mucosal barrier, and, as a consequence, pro-inflammatory and pro-angiogenic products through the gut lining may enter the circulation and directly promote inflammation by activating immune cells and endothelial cells, known triggers in the onset and progression of CVD [11]. Finally, during IBD flares the adhesion of circulating monocytes to the intestinal microvascular endothelial cells, as well as their infiltration and transformation into macrophages occur, in close analogy to what occurs in the early phases of atherosclerosis [12,13]. The occurrence of endothelial dysfunction disrupts the endothelial barrier permeability, which is part of the inflammatory response in the development of CVD. The boosted expression of endothelial adhesion molecules, such as vascular cell adhesion molecule (VCAM)-1, intercellular adhesion molecule (ICAM)-1, and of chemoattractants, is involved in the recruitment of monocytes/macrophages to endothelial cells, which is an obliged step in atherosclerotic development and progression [14].

The Mediterranean diet, characterized by a high intake of vegetables and fruits, exhibits healthy properties and provides significant reduction of chronic disease risk, attributable to the synergistic actions of anti-oxidant and anti-inflammatory compounds [15,16]. Adherence to the Mediterranean diet is low in IBD patients [17]. Therefore, additional or alternative strategies to reduce cardiovascular risk and curb intestinal and systemic inflammation are highly desirable. Grapes are one of the largest fruit crops in the world, and almost all of the production (about 80% of the yield) is used for wine making. Due to the wide availability of grape skins, the utilization of grape by-products has attracted increasing attention for their potential health benefits not only for their antioxidant activity, but also for their antibacterial, anti-inflammatory, and anticarcinogenic properties [18,19]. These biological properties are believed to be due to the functions of polyphenols and dietary fibers still contained in grape pomace after grape fermentation. Among grape pomace compounds with high nutraceutical value [20,21], polyphenols such as phenolic acids, flavanols, proanthocyanidins, flavonols, anthocyanins, and stilbenes are the most interesting due to their antioxidant, anti-inflammatory, anti-neurodegenerative, anti-microbial, anti-cancer, and cardioprotective activities [19,22]. For these reasons, grape pomace could be exploited as a natural source of functional polyphenolic compounds for food or pharmacology use [21,23,24]. The incorporation of these remnants into food products or their use as food supplements could be useful for health promotion and chronic disease prevention, including IBD [25].

Our group has already shown anti-oxidant properties of those polyphenols and polyphenolic extracts from skin pomace in non-intestinal in vitro models. These dietary polyphenols were able to attenuate endothelial dysfunction and reduce leukocyte adhesion to the endothelium [26,27]. Moreover, recent evidence reports grape pomace to improve the gut microbiota by increasing the beneficial bacteria and decreasing the harmful bacteria, as well as reducing the level of pro-inflammatory cytokines [19,28]. However, the biological health potential of grape pomace-derived polyphenols on intestinal and vascular inflammation and the underlying mechanisms of action are not entirely clear.

The aim of the present study was to analyze whether a polyphenol extract of grape marc inhibits intestinal and endothelial inflammation. For this purpose, we used an in vitro inflammatory model consisting of intestinal epithelial cells and endothelial cells that become dysfunctional upon pro-inflammatory stimuli. As intestinal cells, we used Caco-2 cells, which are able to spontaneously differentiate into an enterocyte-like phenotype [29,30]. Differentiated Caco-2 cells were stimulated with pathological concentrations of lipopolysaccharide (LPS) and tumor necrosis factor-α (TNF-α) to mimic the gut inflammatory milieu. The effects of grape pomace extract (GPE) on inflammatory markers in intestinal cells were evaluated, and the underlying mechanism of action was explored assessing intracellular ROS levels and NF-κB activation. Moreover, the crosstalk between intestinal and endothelial cells was investigated by analyzing the transepithelial effect of GPE on endothelial dysfunction. Finally, the leukocyte adhesion to inflamed endothelium was evaluated.

## 2. Materials and Methods

### 2.1. Reagents

Reagents were acquired from various suppliers: cyanidin 3-O-glucoside chloride, rutin (quercetin 3-O-rutinoside), and chlorogenic acid (5-caffeoylquinic acid) were purchased from Extrasynthèse (Genay, France); gallic acid, Folin–Ciocalteu phenol reagent, Trolox ((S)-(-)-6-hydroxy-2,5,7,8 tetramethylchroman-2-carboxylic acid), fluorescein disodium, ABTS (2,2′-azino-bis (3-ethylbenzothiazoline-6-sulfonic acid)), AAPH (2,2′-azobis (2-methyl-propionamide)), acetonitrile, formic acid, ethanol, and organic acids (all HPLC-grade) were acquired from Sigma-Aldrich (St. Louis, MO, USA). Milli-Q water was used (Merck Millipore, Darmstadt, Germany).

### 2.2. Grape Pomace Polyphenolic Extract

Grape pomace (*Vitis vinifera* L., cv Negramaro) was obtained from a winemaking facility (Azienda Agricola Cantele, Guagnano, Lecce, Italy). The wet pomace was dried in an oven at 50 °C until constant weight. Subsequently, the skins were manually recovered from the pomace samples and stored at room temperature in the dark until further processing.

Polyphenol compounds were extracted from a fine powder of grape pomace obtained by freezing the samples in liquid nitrogen and grinding with a blender. The samples (1 g) were treated with 10 mL of methanol/ethanol (80:20, *v*/*v*) in an ultrasound bath (Labsonic Falc, LBS1-H3) at 35 kHz and 88 W for 5 min. Samples were extracted at room temperature for 16 h in the dark under continuous stirring. Extraction mixtures were centrifuged (4000× *g*) for 5 min, the supernatants were collected, and the solvent was evaporated under N_2_ flow. Dried grape pomace extracts (GPE) were solubilized in 70% ethanol and stored at −20 °C until analysis.

### 2.3. High-Performance Liquid Chromatography (HPLC) Characterization of Polyphenols

To quantify the polyphenolic molecules in alcoholic extracts, we performed HPLC analysis using an Agilent-1100 liquid chromatograph equipped with a DAD detector (Agilent 1100 HPLC system, Santa Clara, CA, USA) as described by Gerardi et al. [21]. The chromatographic analysis was performed by comparing each peak retention time with the retention time and UV-visible spectra of external standards.

### 2.4. Folin–Ciocalteu Assay

A rapid method [31] was used to measure the total phenols in alcoholic and water extracts from dried whole and skin pomace in 96-well plates (Corning) using a microplate reader (Infinite 200 Pro, Tecan, Männedorf, Switzerland). Folin–Ciocalteu reagent (1:5, *v*/*v*, 50 µL) and sodium hydroxide solution (0.35 mol/L, 100 µL) were added to each well. The absorbance value at 760 nm was recorded after 5 min incubation. Gallic acid was used to obtain a calibration curve in the range from 2.5 to 40.0 mg/L (R ≥ 0.9997). Gallic acid equivalents (GAE) were used to express the total phenol content of different samples.

### 2.5. Trolox Equivalent Antioxidant Capacity (TEAC) Assay

The TEAC assay was performed as previously reported [21]. The ABTS radical, diluted in PBS (pH 7.4), showed an absorbance value of 0.4 (read at 734 nm). A volume of 200 µL of diluted ABTS was added to 10 µL of extract. Then, the absorbance value was recorder at 734 nm after 6 min using a plate reader (Infinite 200 Pro, Tecan, Männedorf, Switzerland). TEAC values were obtained considering the percentage inhibition at 734 nm with Trolox as standard (0–16 μmol/L were used to obtain a standard curve). TEAC values were expressed as Trolox equivalents (µmol/g).

### 2.6. Oxygen Radical Absorbance Capacity (ORAC) Assay

The ORAC procedure was accomplished as per Gerardi et al. [21]. Briefly, the reaction was performed using a 96-well plate reader (Infinite 200 Pro, Tecan, Männedorf, Switzerland) in a 75 mmol/L phosphate buffer (pH 7.4), in 200 µL final reaction volume. The mixture of dried grape pomace extracts (20 µL) and fluorescein solutions (120 µL; 70 nmol/L) was heated at 37 °C for 15 min. Then, 2,2′-azobis-(2-methylpropionamidine) dihydrochloride was added, and the fluorescence was recorded (excitation and emission wavelengths of 485 and 527 nm, respectively) every minute for 60 min. A blank using phosphate buffer instead of the sample was carried out in each assay and all the reaction mixtures were assessed in triplicate. Decay curves (fluorescence intensity vs. time) were recorded and the net area under the curve was calculated by subtracting the blank value from that of sample or standard. The antioxidant capacity was quantified using the antioxidant Trolox as a standard. Final ORAC values were expressed as Trolox equivalents (µmol/g) of dried grape pomace extract.

### 2.7. Cell Cultures

The human colorectal adenocarcinoma-derived intestinal epithelial cell line Caco-2 was obtained from the American Tissue Culture Collection (Rockville, MD, USA) and cultured in Dulbecco’s modified Eagle’s medium (DMEM) supplemented with 10% heat-inactivated fetal bovine serum (FBS), 2 mmol/L glutamine, 1% non-essential amino acids, 100 U/mL penicillin, and 100 µg/mL streptomycin, in monolayers at 37 °C in a humidified atmosphere of 5% CO_2_ as previously described [32]. The human microvascular endothelial cell line (HMEC-1), obtained from Dr. Thomas J. Lawley, was cultured in MCBD 131 supplemented with 10% FBS, 2 mmol/L glutamine, and 100 U/mL penicillin and 100 µg/mL streptomycin, and grown at 37 °C in a humidity-controlled 5% CO_2_ cell culture incubator, as previously described [33,34]. Human monocytoid THP-1 cells were obtained from the American Tissue Culture Collection (Rockville, MD, USA) and maintained in RPMI 1640 medium supplemented with 10% FBS, 2 mmol/L glutamine, and 100 U/mL penicillin and 100 µg/mL streptomycin in a 5% CO_2_ humidified atmosphere at 37 °C.

### 2.8. Cell Viability

Cell viability was determined by a 3-(4,5-dimethylthiazol-2-yl)-2,5-diphenyl tetrazolium bromide (MTT) assay, which is a commonly used method to evaluate cell survival, on the basis of the ability of viable cells to convert MTT, a soluble tetrazolium salt, into an insoluble formazan precipitate, which is then quantitated spectrophotometrically [35]. Briefly, cells were seeded on the 96-well culture plates and treated with various concentration of GPE (1, 5, 10, and 25 μg/mL GAE) for 24 h. Then, cells were incubated with MTT (0.5 mg/mL) for 4 h. After that, the formazan products were dissolved in isopropanol and absorbance was measured at 540 nm using a microplate reader. The results were expressed as percentage compared to untreated cells.

### 2.9. Treatments of Caco-2 Cell Monolayers

For the experiments, Caco-2 cells were grown on 12-well plates for 21 days to obtain spontaneous differentiation towards enterocyte-like cells, replacing the medium every 2 days, as previously described [32]. After differentiation, cell culture medium was shifted to 3% FBS, and then Caco-2 cells were treated with different concentrations of GPE (1, 5, and 10 μg/mL GAE) for 2 h; after that, cells were stimulated with LPS 10 µg/mL and TNF-α 10 ng/mL for 16 h (Figure 1A). Cells exposed to LPS plus TNF-α only were considered inflamed controls (positive), whereas those without any treatment were considered negative controls. After incubation, culture media were collected in sterile microtubes and the cells were lysed with TRIzol reagent solution (Thermo Fisher Scientific, Waltham, MA, USA) or specific lysis buffer to obtain RNA or nuclear protein, respectively. Culture media, as well as cell lysates, were stored at −80 °C until analysis. Three biological replicates were performed.

### 2.10. Caco-2/HMEC-1 Co-Culture System and Treatments

For the co-culture experiments, Caco-2 cells were grown on semipermeable filters over 21 days to differentiate and develop an enterocyte-like phenotype, as previously described [36,37,38]. Then, Caco-2 cells were moved to 6- or 12-well plates, containing HMEC-1 cells on the bottom compartment. GPE (1, 5, and 10 μg/mL GAE) was added on the apical compartment for 2 h. Afterwards, LPS (10 μg/mL) and TNF-α (10 ng/mL) were applied on the apical compartment, and TNF-α (10 ng/mL) on the basolateral compartment for 16 h (Figure 1B).

### 2.11. Leukocyte Adhesion Assay

After the above-described treatments, trans-well inserts containing Caco-2 monolayers were removed and HMEC-1 was washed with DPBS before leukocyte assay (Figure 1C). THP-1 cells were labeled with 1 μmol/L calcein AM (Molecular Probes, a brand of Thermo Fisher Scientific, Waltham, MA, USA) for 30 min in DMEM medium containing 3% FBS. In a co-culture system, labelled THP-1 cells were seeded at 5 × 10^5^ cell density onto the HMEC-1 monolayer and incubated under rotating conditions (63 rpm) at 21 °C, as previously described [26]. After a gentle wash to remove unattached cells, the adherent THP-1 cells were observed under fluorescence microscope. Alternatively, the fluorescence intensity was measured in a microplate reader with an excitation/emission wavelength of 485/530 nm.

### 2.12. Detection of Endothelial Cell Surface Molecules

After the treatments described above, the trans-well inserts containing Caco-2 monolayers were removed, and HMEC-1 was used for the detection of endothelial cell surface molecules VCAM-1 and ICAM-1 by cell surface enzyme immunoassays (EIA), as described previously [26].

### 2.13. Measure of Inflammation Marker Release

The conditioned media from Caco-2 monolayers and the basolateral compartment of Caco-2/HMEC-1 co-culture were collected, and the levels of secreted IL-6, MCP-1, MMP-9, and MMP-2 were determined using the corresponding enzyme-linked immunosorbent assay (ELISA) kits, according to the manufacturer’s instructions.

### 2.14. RNA Isolation and Real-Time Quantitative PCR

Total RNA was extracted from Caco-2 monolayers and HMEC-1 in co-culture systems, using the TRIzol reagent (Thermo Fisher Scientific, Waltham, MA, USA), and quantified spectrophotometrically. Total RNA (1 μg) was converted into first-strand cDNA using the High-Capacity cDNA Reverse Transcription Kit (Applied Biosystems, Monza, Italy). Quantitative RT-PCR was performed in CFX384 Touch Real-Time PCR Detection System (Bio-Rad Laboratories, Milan, Italy) using a SYBR Green PCR Master Mix (Bio-Rad Laboratories, Milan, Italy) and the synthesized primers (Thermo Fisher Scientific, Waltham, MA, USA) reported in Table 1. All reactions were assessed in triplicate. The mRNA quantity was calculated by comparative critical threshold method [38]. As endogenous reference ribosomal RNA 18 S and GAPDH was simultaneously quantified for each sample, and the data were normalized accordingly. Results are expressed as fold increase relative to unstimulated control (=1).

### 2.15. Preparation of Nuclear Protein Extracts and NF-κB Activation Assay

Caco-2 cells were treated with GPE for 2 h and stimulated with LPS and TNF-α for an additional 2 h, after which nuclear proteins were purified by using a kit from Active Motif (Carlsbad, CA, USA). Briefly, cell monolayers were collected in ice-cold PBS containing phosphatase inhibitors, then centrifuged at 300× *g* for 5 min. Pellets were resuspended in a hypotonic buffer and centrifuged at 14,000× *g* for 30 s. Nuclear proteins were solubilized in lysis buffer containing proteasome inhibitors, and protein concentrations were determined by Bio-Rad protein assay (Bio-Rad Laboratories, Milan, Italy). The DNA-binding activity of NF-κB (p65) was quantitated using the Active Motif’s ELISA-based “TransAM kit” following the manufacturer’s instructions.

### 2.16. Detection of Intracellular ROS Production

Cellular ROS levels were assessed using a carboxy-2,7-dichlorofluorescein diacetate (CM-H2DCFDA) probe, as described previously [34]. CM-H2DCFDA is hydrolyzed in the cytosol to form the DCFH carboxylate anion. Oxidation results in the formation of fluorescent DCF, which is maximally excited at 495 nm and emits at 520 nm. Differentiated Caco-2 cells were incubated with GPE for 2 h and stimulated with LPS and TNF-α for an additional 2 h, and, after that, loaded with the probe CM-H2DCFDA (10 μmol/L) for 45 min at 37 °C in the dark. Following CM-H2DCFDA incubation, monolayers were gently washed twice in PBS; then, phenol red-free medium was added, and fluorescence was monitored by spectrofluorimetric analysis.

### 2.17. Statistical Analysis

Data are expressed as mean ± standard deviation (SD) of at least three independent experiments. Differences between two groups were determined by unpaired Student’s *t*-test. Multiple comparisons were performed by one-way analysis of variance (ANOVA), and individual differences were then tested by the Fisher’s protected least-significant difference test after the demonstration of significant inter-group differences by ANOVA. A *p* value < 0.05 was considered to be statistically significant.

## 3. Results

### 3.1. Total Phenolic Content and Antioxidant Potential of GPE

In the present study, we used a polyphenolic extract from the grape pomace of the Negramaro cv (GPE), by applying an eco-sustainable and safe strategy previously developed by our research group [21]. Results for GPE antioxidant activity (TEAC and ORAC), total phenolic (TP) content, and polyphenolic profile are reported in Table 2 and Figure 2. We found that GPE exhibited an antioxidant activity and phenolic content comparable to our previous data [21]. The main components of GPE were oenin, epicatechin, catechin, gallic acid, quercetin, and quercetin-3-glucoside (Figure 2). Oenin and epicatechin were found to be the most representative constituents (with resulting amounts of 4.9 and 3.7 mg/g, respectively) (Figure 2).

### 3.2. GPE Prevents the Stimulated Expression of Proinflammatory Mediators in Caco-2 Cells

We next investigated the effects of GPE on intestinal inflammatory parameters using Caco-2 cell monolayers. We preliminarily assessed the effects of GPE on cell viability using the MTT assay. To this aim, Caco-2 cells were exposed to increasing concentrations (1, 5, 10, and 25 µg/mL) of GPE for 24 h (Figure 3). The MTT assay results suggest that concentrations equal to 25 µg/mL started to be toxic for Caco-2 cells (toxicity >20%) compared to the vehicle control. On the basis of these results, in the following experiments, we used GPE concentrations of 10 µg/mL or less, which were not cytotoxic and could represent plausible concentrations in the human gut [39].

To evaluate the anti-inflammatory effects of GPE, differentiated Caco-2 monolayers were pre-treated with GPE for 2 h and stimulated with LPS and TNF-α to mimic the intestinal inflammatory milieu. Under these experimental conditions, the production of the cytokine IL-6 and the chemokine MCP-1 in Caco-2 cells was analyzed by ELISA assay. We observed that in Caco-2 cells, LPS/TNF-α stimulation induced a significant release of the proinflammatory mediators IL-6 and MCP-1 as compared with the unstimulated control cells (Figure 4). The pre-treatment with GPE significantly reduced the production of both proinflammatory mediators in a concentration-dependent manner. GPE at 10 µg/mL brought the production of IL-6 and MCP-1 back to control values (Figure 4A,B). To verify the mechanisms underlying the reduced release of these proinflammatory molecules, we investigated the effects of GPE on IL-6 and MCP-1 gene expression. For this purpose, Caco-2 cells were incubated with GPE for 2 h before stimulation with LPS/TNF-α for 16 h, and then mRNA levels were measured by quantitative RT-PCR (Figure 4C,D). Consistent with decreased protein release, the pretreatment of Caco-2 cells with GPE also lowered the mRNA levels of IL-6 and MCP-1 (Figure 4C,D).

Furthermore, we also investigated the effects of GPE on the expression of other proinflammatory mediators that play a key role in intestinal inflammation, such as the matrix metalloproteinases. GPE pre-treatment of Caco-2 significantly reduced the LPS/TNF-α-induced release and expression of MMP-9 and MMP-2 in a concentration-dependent manner (Figure 5A,B). Since matrix metalloproteinase inhibitors (TIMP) regulate MMP activity, we also analyzed the TIMP-1 and TIMP-2 gene expression in LPS/TNF-α challenged Caco-2 cells. Although LPS/TNF-α stimulation did not alter the expression of TIMP-1 and TIMP-2, cell exposure to 10 μg/mL GPE produced a significant increase in the mRNA levels of both TIMP-1 and TIMP-2 (Figure 5C). 

Moreover, GPE was able to significantly reduce the LPS/TNF-α-induced mRNA levels of several inflammatory markers including the cytokines IL-1β and TNF-α, the chemokines CXCL-10 and M-CSF, the inflammatory enzyme cyclooxygenase-2 (COX-2), and the adhesion molecules VCAM-1 and ICAM-1 (Table 3). These data suggest a possible protective effect of grape pomace polyphenols in the gut by inhibiting the expression and release of key inflammatory mediators.

### 3.3. GPE Reduced the Stimulated Activation of NF-κB and the Intracellular ROS Levels in Caco-2 Cells

The expression of inflammatory genes is mainly regulated by NF-κB signaling pathways. Under resting conditions, NF-κB is relegated in cytoplasm in an inactive form; inflammatory stimuli activate NF-κB, which translocates to the nucleus, where it binds to promoters of target genes, such as those codifying for proinflammatory proteins [40]. In Caco-2 cells, we verified the effect of GPE on LPS/TNF-α-stimulated NF-κB activation by an ELISA-based method, which measures the DNA-binding activity of the NF-κB subunit p65. The activation of NF-κB was evidenced by the increased levels of p65 subunit in the nuclear extracts of Caco-2 cells treated with LPS/TNF-α for 2 h, whereas GPE pre-treatment prevented, in a concentration-dependent manner, LPS/TNF-α-induced nuclear translocation of p65 (Figure 6A). These results suggest that GPE is able to modulate the activation of NF-κB in response to inflammatory stimuli in intestinal epithelial cells.

Changes in the cellular redox balance may mediate the activation of NF-κB, which can be inhibited by treatment with antioxidants [41]. Because GPE showed strong antioxidant potential in cell-free systems (Table 2), we examined its antioxidant capacity in Caco-2 cells triggered with LPS/TNF-α. The LPS/TNF-α stimulation of Caco-2 cells produced a significant increase in the intracellular levels of ROS, as assessed by the ROS-sensitive carboxy-H2DCFDA probe. Cell treatment with GPE, before LPS/TNF-α stimulation, significantly reduced ROS levels in a concentration-dependent manner (Figure 6B,C), indicating meaningful intracellular antioxidant effects by GPE in our experimental conditions.

### 3.4. GPE Inhibited Endothelial Activation and Monocyte Adhesion in Intestinal Epithelial/Endothelial Co-Culture Model

To mimic intestinal epithelial–endothelial cell interactions in vitro, we used a co-culture system that places endothelial cells in close proximity, but not in contact, to intestinal epithelial cells. In this model, we evaluated if intestinal anti-inflammatory action of GPE was associated with improved endothelial function, assessed as endothelial activation and monocyte recruitment.

To this aim, differentiated Caco-2 monolayers, grown on the upper side of semipermeable trans-well inserts, were placed in close proximity to HMEC-1 grown on the bottom of the wells (see Figure 1 for details). GPE was added on the apical compartment for 2 h; then, LPS and TNF-α were applied on the apical side, and TNF-α on the basolateral side. After 16 h of inflammatory stimulation, the release of IL-6 and MCP-1 in the conditioned media of the basolateral side was evaluated by ELISA.

The results showed that inflammatory stimuli significantly intensified the release of IL-6 and MCP-1 in co-culture system as compared with unstimulated cells (Figure 7A,B). Pre-treatment of Caco-2 with GPE dose-dependently reduced the release of both inflammatory mediators in co-culture system (Figure 7A,B), confirming and strengthening the data obtained in Caco-2 monoculture.

To analyze the specific contribution of endothelial cells in the release of IL-6 and MCP-1 in Caco-2/HMEC-1 co-culture model, mRNA levels of both inflammatory mediators were analyzed in HMEC-1. The results showed that inflammatory stimulation induced a significant increase in the mRNA levels of IL-6 and MCP-1 that was reduced in a concentration-dependent manner by pre-treatment with GPE (Figure 7C,D).

In co-cultured endothelial cells, pro-inflammatory stimuli significantly induced the expression of VCAM-1 and ICAM-1 at the protein and mRNA levels, as evaluated by cell surface EIA and quantitative RT-PCR, respectively (Figure 8A,B). The results showed that GPE were able to downregulate the endothelial cell adhesion molecule expression in a concentration-dependent manner (Figure 8A,B), without affecting the expression of the constitutive endothelial surface antigen E1/1 (data not shown).

Endothelial expression of adhesion molecules is responsible for leukocyte adhesion to vascular endothelium and migration into the subendothelial space. In order to confirm the inhibitory activity of GPE on endothelial activation in Caco-2/HMEC-1 co-culture, the adhesion of human leukocytes THP-1 on HMEC-1 was performed as described in Section 2.11. As shown in Figure 8C, the number of leukocytes adhering to the stimulated endothelial cells was higher than that observed in unstimulated control. Leukocyte adhesion appeared to be reduced in a concentration-dependent manner by Caco-2 pre-treatment with GPE, demonstrating that grape pomace polyphenols were able to reduce intestinal inflammation and prevent the consequent endothelial activation and leukocyte adhesion.

## 4. Discussion

Agroindustrial waste from food production is now reviewed as a valuable product due to its high content of phytochemicals with purported health benefits. During winemaking, about 20–25% of the grape weight remains as waste material [42]. Although mainly used in the past as livestock feed, soil fertilizer, or fuel, they can be also exploited in the production of new nutraceuticals and functional foods to promote health and prevent disease [25]. Grape pomace composes mainly of skins and seeds, which, due to poor extraction during winemaking, still contain high levels of phytochemical compounds, especially polyphenols [43], which are the main secondary metabolites of plants and an integral part of the human diet. It has been reported that grape pomace contains a great variety of polyphenols including anthocyanins, catechins, flavonols, alcohols, stilbenes, and benzoic (gallic, protocatechuic, 4-hydroxybenzoic) and cinnamic (p-coumaric) acids [44,45]. In the present study, we confirm our previous findings regarding the polyphenolic profile of Negramaro grape pomace [21], showing that the main components found in GPE were active polyphenols such as oenin, epicatechin, catechin, quercetin and quercetin-3-glucoside, gallic acid, caftaric acid, and resveratrol, which are responsible for GPE antioxidant activity. The intestinal epithelium is constantly exposed to dietary phenolic compounds, which are poorly absorbed and reach high concentrations in the lumen, where they can exert health benefits by protecting the intestinal mucosa from oxidative damage and helping to improve antioxidant capacity [46]. A continuous crosstalk between the different cell types present in the gut, including endothelial cells, influences the inflammatory process and the immunological response of the gut, which is crucial for preserving gut health and functioning as well as for preventing CVD [47,48]. A dysregulation of intestinal inflammation can lead to severe gut disorders such as IBD, which is characterized by overproduction of several inflammatory mediators, including TNF-α and LPS, implicated in the initiation and flares of IBD, as well as in systemic inflammatory reaction [49,50]. In vivo and in vitro studies have shown that polyphenols are effective in preventing and relieving IBD symptoms, regulating the intestinal ecosystem, and reducing the level of pro-inflammatory cytokines [46,51,52,53]. Since nutritional interventions based on high polyphenol intake have a lower toxic effect than pharmaceutical approaches [54], ongoing efforts are aimed at developing new dietary polyphenol-based strategies targeting oxidative stress and inflammatory signaling events. The aim of the present study was therefore to explore the effect of polyphenolic extract of grape pomace on intestinal inflammation and the transepithelial impact of GPE compounds on endothelial dysfunction.

Since GPE could act directly on intestinal epithelial cells before crossing the blood–intestinal barrier, we first investigated the potential anti-inflammatory properties of GPE in Caco-2 cell monolayers, a model of human intestinal epithelium. These cells, derived from a human colon adenocarcinoma, once in culture undergo a process of spontaneous differentiation into normal mature enterocytes [29,30]. To try and mimic the intestinal inflammatory milieu detectable during IBD, we exposed Caco-2 cells to TNF-α and LPS. Our results in Caco-2 cells showed that LPS and TNF-α activated intracellular cascades leading to increased transcriptional activity and secretion of the cytokines IL-6 and MCP-1 and the matrix metalloproteinases MMP-9 and MMP-2. We also observed an overall boosted gene expression of other inflammatory markers, such as the cytokines IL-1β and TNF-α, the chemokines CXCL-10 and M-CSF, the pro-inflammatory enzyme COX-2, and the adhesion molecules VCAM-1 and ICAM-1. In our experimental model, the anti-inflammatory property of GPE against LPS/TNF-α stimulation was demonstrated in terms of IL-6 and MCP-1 decrease, at protein and mRNA levels, occurring in a concentration-dependent manner. This finding can be relevant to counteract gut inflammation in IBD. Indeed, IL-6 and MCP-1 are upregulated in IBD patients and play an important role in mucosal immune responses [55,56]. The 10 μg/mL concentration of GPE provided the most consistent results across all measures, as it was also associated with the highest suppression of pro-inflammatory mediators including IL-1β and TNF-α, CXCL-10 and M-CSF, COX-2, and VCAM-1 and ICAM-1. GPE concentrations equal to or less than 10 µg/mL, as used in this study, were not toxic in Caco-2 cells. These values, on the other hand, represent biologically relevant concentrations in the human gut lumen [39], where polyphenols may reach high concentrations after dietary consumption [46]. Moreover, GPE pre-treatment significantly reduced the LPS/TNF-α-induced release and expression of MMP-9 and MMP-2 in Caco-2 cells, in a concentration-dependent manner, while it augmented the mRNA levels of metalloproteinase inhibitors TIMP-1 and TIMP-2. Extracellular MMPs are essential factors involved in the development, modification, and healing of inflammatory lesions. MMP-9 is suggested to be the key factor determining mucous membrane damage in IBD [57,58]; thus, it can be a potential therapeutic target. Indeed, therapeutic strategies with infliximab (anti-TNF-α) enhance mucosal healing by reducing MMP-9 activity through an increased expression of TIMP-1, the most potent natural inhibitor of MMP-9 [59]. To the best of our knowledge, these results show for the first time that polyphenolic extracts from grape pomace can improve gut health by restoring the balance between MMP-9/2 and TIMP 1/2, which is altered by the intestinal inflammatory environment. In this study, we also investigated the molecular mechanism of GPE action in inflamed Caco-2 monolayer. We found that GPE was able to revert the LPS/TNF-α-induced activation of NF-κB, which plays a pivotal role during intestinal inflammatory responses. Thus, the ability of GPE to counteract the activation of NF-kB signaling pathway may explain the anti-inflammatory activities of wine pomace polyphenols at the intestinal level. This effect was associated with decreased intracellular ROS content in inflamed Caco-2 cells, which was related to the antioxidant capacity of GPE. Intestine is a key source of ROS due to necessary exposure to foreign substances and microbial pathogens. Disproportionate generation and long-term exposure to ROS lead to various inflammatory intestinal diseases, such as IBD [49]. GPE, by dumping intracellular ROS, may protect gut health and mitigate oxidative stress-induced damage. Our results confirm and extend previous studies showing significant protection by the polyphenolic components of grape marc against inflammation, oxidative stress [60,61], and NF-κB activation [62,63], thus restoring tight junction barrier function in Caco-2 colon cells [64]. Similar effects have been also reported for pure polyphenols. In particular, malvidin-3-glucoside has been shown to have anti-inflammatory properties in a murine colitis model through the modulation of the integrity of the colon epithelium, gut microbiota, and gut metabolism [65,66]. Catechins and epicatechins influenced cellular response to oxidative stress and exerted intestinal anti-inflammatory properties by modulating several cell signaling pathways, such as NF-κB, mitogen-activated protein kinases (MAPKs), and transcription factor nuclear factor (erythroid-derived 2)-like 2 (Nrf2) [67,68]. Similarly, kaempferol counteracted gut inflammation by inhibiting the activation of the NF-κB signaling pathway in intestinal epithelial–endothelial co-culture model [69]. In intestinal in vitro and in vivo models, both quercetin and its metabolites, including rutin, exhibited anti-inflammatory properties by inhibiting the activity of NF-κB and reducing the production of several proinflammatory mediators [70,71,72]. Although the in vitro effects of resveratrol on intestinal inflammation are contradictory [70], animal studies have shown that resveratrol can reduce the severity of intestinal inflammation in IBD models by downregulating several intestinal immunity mediators and affecting key components of the inflammatory cascade, mainly the inhibition of NF-κB activation and the attenuation of reactive species production [73]. Moreover, findings showed that gallic acid protected Caco-2 cells against oxidative stress [60] and inhibited inflammation in dextran sulfate sodium-induced colitis in mice through the suppression of p65-NF-κB [74]. In vitro digestion studies revealed that phenolic acids, in addition to quercetin, were the polyphenols most resistant to digestion, and could be the most relevant to explain the biological activity of digested foods [75].

Overall, our data on GPE are in accordance with previous in vitro and in vivo results on the intestinal anti-oxidant and anti-inflammatory signaling pathways of pure polyphenols, although further investigations are necessary to fully clarify the underlying mechanisms of action.

Since oxidative stress and immune response are common mechanisms in both IBD and CVD, our findings suggest that GPE could be an effective strategy to improve gut health and prevent CVD. Recent evidence shows that gut inflammation in IBD is an independent risk factor for endothelial dysfunction and related CVD. In this contest, to assess the gut–vascular axis, we analyzed the impact of transepithelial GPE on inflamed endothelial cells and leukocyte adhesion. We tested the biological activity of GPE in a well-validated co-culture model of intestinal epithelial/endothelial cells [30,36,37,38,76,77], analyzing the transepithelial effects in HMEC-1 cells under inflamed conditions. We found, for the first time, that GPE pre-treatment of Caco-2 in co-culture system reduced endothelial activation in terms of release and expression of IL-6 and MCP-1, as well as the endothelial expression of VCAM-1 and ICAM-1. As a functional implication, GPE anti-inflammatory effects were related with a reduced endothelial leukocyte recruitment in response to inflammatory stimuli, which can suggest an additional protective effect of GPE in attenuating both vascular and intestinal inflammation with potential joint beneficial impact on IBD and CVD. Our findings regarding the multiple anti-inflammatory effects at the intestinal and endothelial levels extend previous studies showing that grape polyphenols, including resveratrol, kaempferol, or anthocyanins, exert a protective effect in endothelial cells against oxidative stress and inflammation, which are closely connected with cardiovascular diseases [37,38,69]. 

The beneficial actions of polyphenols are largely dependent on their bioavailability at the target tissues [70]. This bioavailability depends on the absorption of polyphenols at the gastrointestinal level, as well as on their metabolism [70]. The intestinal absorption of polyphenols depends on their chemical structures; some glycoside forms, mainly glucosides, interact with proteins, including the glucose transporters, the cytosolic b-glucosidase, or the membrane-bound lactase-phlorizin hydrolase, while aglycone forms seem to undergo passive diffusion only [78,79]. Typically, polyphenolic compounds can be subjected to catabolism by the intestinal microbiota, which breaks down the polyphenolic compound, producing smaller molecules, including phenolic acids and hydroxycinnamates, which are absorbed by the colon. Before passage into the blood stream, absorbed compounds can undergo some degree of phase II metabolism, forming sulfate, glucuronide, and/or methylated metabolites through specific metabolic enzymes [79]. In particular, resveratrol undergoes excessive metabolism by intestinal epithelial cells, and very low amounts of unconjugated resveratrol circulate in the blood, while sulphate and glucuronate conjugates are present in higher amounts [80,81]. These conjugates have also been found in studies conducted in Caco-2 cells treated with resveratrol or kaempferol [38,82], thus confirming the value of using an intestinal cell culture in the study of polyphenol absorption and biological effects. Other previous data showed that dietary polyphenols, including caffeic acid, gallic acid, quercetin, rutin, and resveratrol, were poorly absorbed by Caco-2 cells, and their transepithelial transports occurred mainly by passive diffusion [83]. A recent study reported that anthocyanins from red wine, including malvidin-3-glucoside, were transported through Caco-2 cells [84]. In vivo study reported that malvidin-3-glucoside was found in plasma and urine after ingestion of anthocyanin-containing beverages, and its plasma concentration correlated with the amount of the anthocyanin ingested [85]. 

Overall, our research shows that GPE polyphenols are biologically active and effective in counteracting excessive inflammation by acting on both intestinal and endothelial cells. Therefore, GPE could be exploited as a “green source” of nutraceuticals for food supplement or components of functional food to improve the nutritional status of subjects with IBD, prevent intestinal inflammation, and assist in therapeutic strategies.

## 5. Conclusions

In conclusion, our results suggest that protective effects of grape pomace polyphenols may begin in the gut, where they counteract the excessive inflammatory response by exhibiting ROS scavenging ability and inhibiting the activation of redox-sensitive transcription factors. We show for the first time that physiological concentrations of grape pomace polyphenols ameliorate the inflammatory response in intestinal cells by downregulating the expression of pro-inflammatory cytokines, chemokines, adhesion molecules, and matrix metalloproteinases, which are all crucial mediators of the oxidative and inflammatory process in IBD. Furthermore, GPE exhibit bioactivity on inflamed endothelial cells by reducing endothelial activation and subsequent leukocyte adhesion. Although it is difficult to directly transfer these data to the in vivo conditions, our results suggest that grape pomace is a natural source of polyphenols with multiple healthy properties that could contribute to the development of new ingredients and nutraceuticals able to relieve chronic gut inflammatory diseases and improve vascular endothelial functions.

## Figures and Tables

**Figure 1 nutrients-14-01175-f001:**
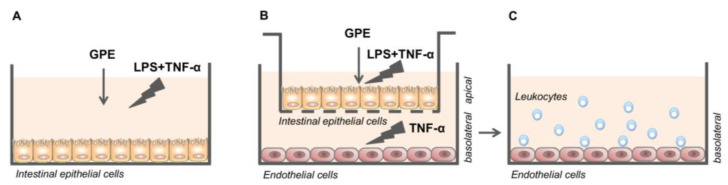
Experimental design. Caco-2 cell monolayers (**A**); Caco-2/HMEC-1 co-culture system (**B**) and subsequent adhesion of human leukocytes to HMEC-1 (**C**). GPE: Grape pomace extract; LPS: Lipopolysaccharide; TNF-α: Tumor necrosis factor-α.

**Figure 2 nutrients-14-01175-f002:**
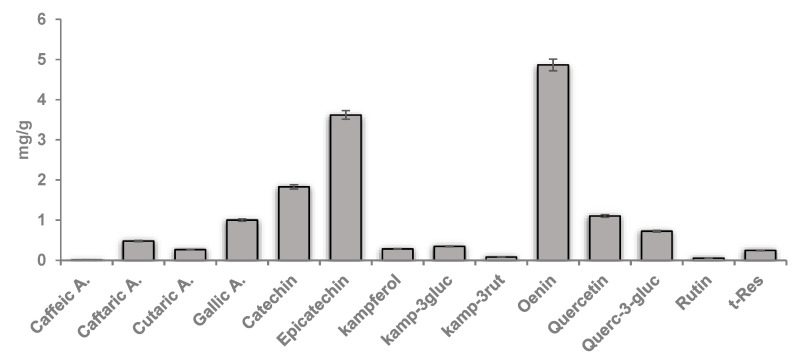
Characterization of different phenolic compounds occurring in Negramaro grape pomace extract (GPE) suspended in ethanol 70% and used for bio-activity assays.

**Figure 3 nutrients-14-01175-f003:**
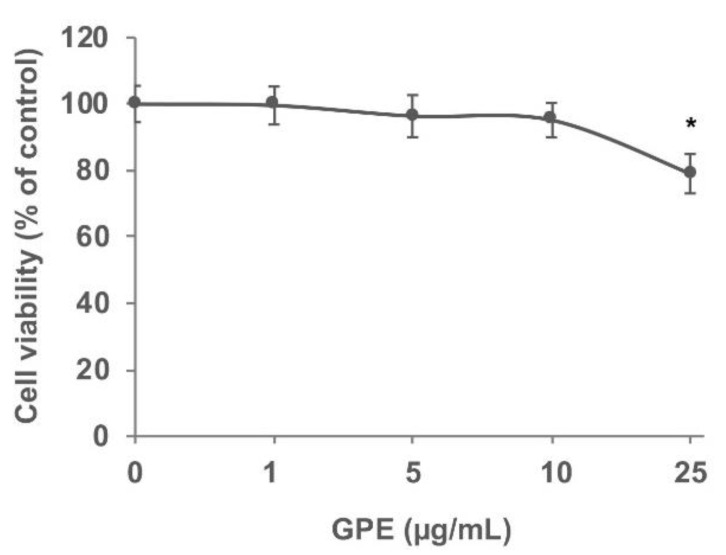
GPE effect on Caco-2 cell viability. Caco-2 cells were seeded into a 96-well plate and treated with increasing concentrations of GPE (1, 5, 10, and 25 µg/mL) for 24 h. Cell viability was determined using MTT assay. Data are reported as percentage of untreated control cells (mean ± SD), and they are representative of three independent experiments. * *p* < 0.05 versus untreated control.

**Figure 4 nutrients-14-01175-f004:**
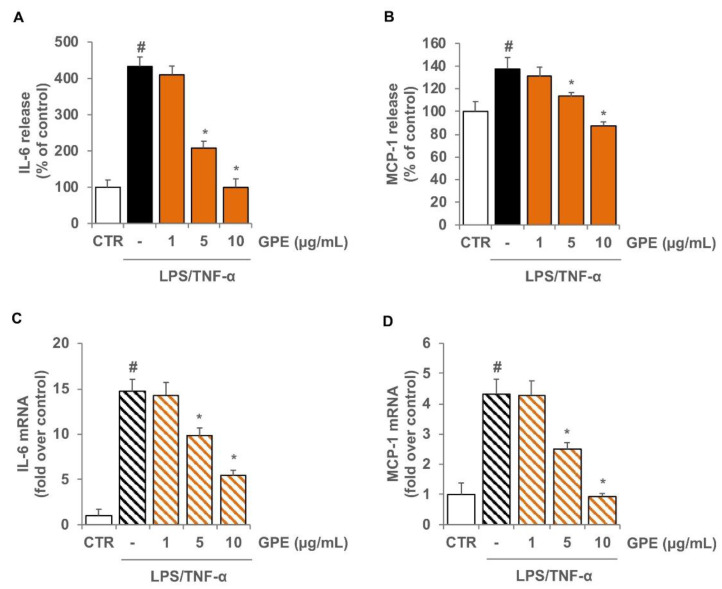
Inhibitory effects of GPE on the expression of IL-6 and MCP-1 in inflamed Caco-2 cells. Differentiated Caco-2 cells were pre-treated with increasing concentrations of GPE (1, 5, and 10 µg/mL) for 2 h, followed by incubation with LPS (10 μg/mL) and TNF-α (10 ng/mL) for 4 h (**C**,**D**) or 24 h (**A**,**B**). IL-6 and MCP-1 release in culture medium was analyzed by ELISA assay. Results are shown as percentage of unstimulated control (mean ± SD). IL-6 and MCP-1 mRNA levels were determined by quantitative RT-PCR and are expressed as fold over unstimulated control (mean ± SD). The white histograms correspond to the untreated control (CTR), while the colored histograms correspond to the different treated groups (black for LPS/TNF-α and orange for GPE+LPS/TNF-α). Data are representative of three independent experiments. # *p* < 0.01 versus control; * *p* < 0.05 versus LPS/TNF-α alone.

**Figure 5 nutrients-14-01175-f005:**
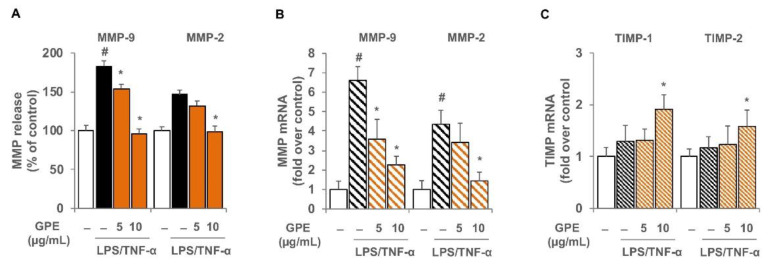
GPE effects on the expression of matrix metalloproteinases and their inhibitors in inflamed Caco-2 cells. Differentiated Caco-2 cells were pre-treated with various concentrations of GPE (5 and 10 µg/mL) for 2 h, followed by incubation with LPS (10 μg/mL) and TNF-α (10 ng/mL) for 16 h (**B**,**C**) or 24 h (**A**). MMP-9 and MMP-2 release in culture medium was analyzed by ELISA assay. Results are shown as percentage of unstimulated control (mean ± SD). MMP-9 and MMP-2 (**B**) as well as TIMP-1 and TIMP-2 (**C**) mRNA levels were determined by quantitative RT-PCR and are expressed as fold over unstimulated control (mean ± SD). The white histograms correspond to the untreated control, while the colored histograms correspond to the different treated groups (black for LPS/TNF-α and orange for GPE+LPS/TNF-α). Data are representative of three independent experiments. # *p* < 0.01 versus control. * *p* < 0.05 versus LPS/TNF-α alone.

**Figure 6 nutrients-14-01175-f006:**
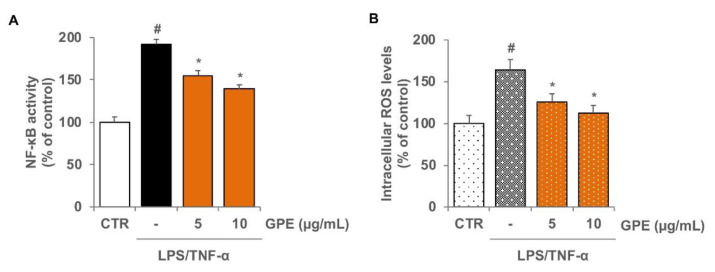
Inhibitory effects of GPE on NF-κB activation and intracellular ROS levels in inflamed Caco-2 cells. Differentiated Caco-2 cells were pre-treated with various concentrations of GPE (5 and 10 µg/mL) for 2 h, followed by incubation with LPS (10 μg/mL) and TNF-α (10 ng/mL) for 2 h. NF-κB activation was assessed in nuclear proteins by an ELISA-based method measuring the DNA-binding activity of NF-κB (**A**). Intracellular ROS were analyzed by using carboxy-H2DCFDA staining by fluorescence plate reader (**B**). The white histograms correspond to the untreated control (CTR), while the colored histograms correspond to the different treated groups (black for LPS/TNF-α and orange for GPE+LPS/TNF-α). Each experiment was executed in triplicate. Data are reported as unstimulated control percentage (mean ± SD). # *p* < 0.01 versus control; * *p* < 0.05 versus LPS/TNF-α alone.

**Figure 7 nutrients-14-01175-f007:**
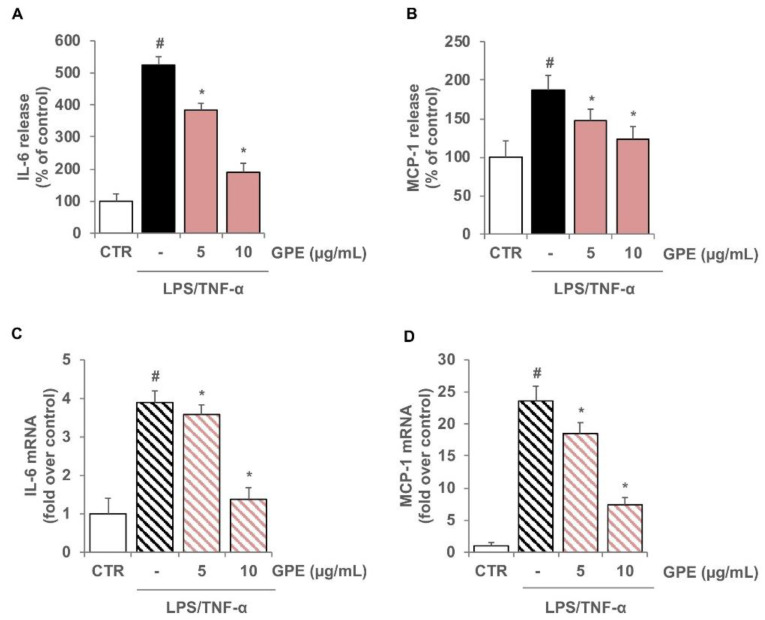
Inhibitory effects of GPE on IL-6- and MCP-1-stimulated expression in a Caco-2/HMEC-1 co-culture model. Differentiated Caco-2 monolayers were grown on the upper side of the inserts and placed in proximity to HMEC-1 grown on the bottom of the wells. GPE (5 and 10 μg/mL) was added on the apical compartment for 2 h, after which LPS (10 μg/mL) and TNF-α (10 ng/mL) were applied on the apical compartment and TNF-α (10 ng/mL) on the basolateral compartment for 16 h (**C**,**D**) or 24 h (**A**,**B**). The protein release of IL-6 and MCP-1 was assessed in basolateral culture medium by ELISA assay. Results are shown as percentage of unstimulated control (mean ± SD). In endothelial cells, IL-6 and MCP-1 mRNA levels were measured by quantitative RT-PCR and are expressed as fold over unstimulated control (mean ± SD). The white histograms correspond to the untreated control (CTR), while the colored histograms correspond to the different treated groups (black for LPS/TNF-α and pink for GPE+LPS/TNF-α). Data are representative of three independent experiments. # *p* < 0.01 versus control; * *p* < 0.05 versus LPS/TNF-α alone.

**Figure 8 nutrients-14-01175-f008:**
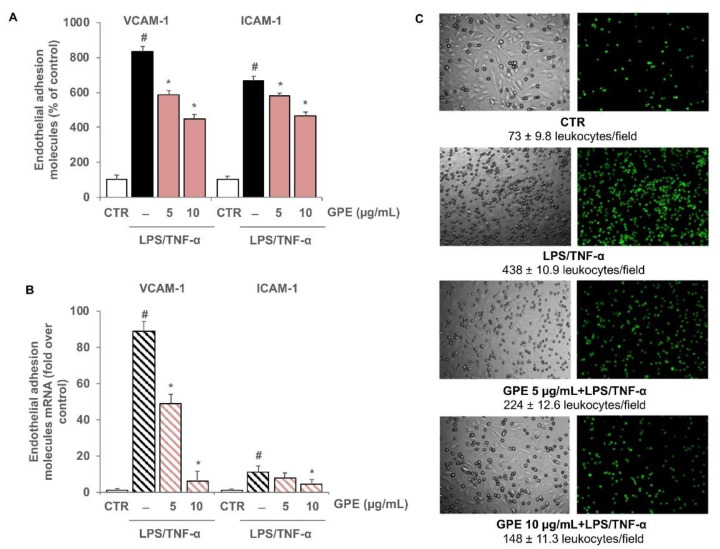
Inhibitory effects of GPE on the stimulated expression of endothelial adhesion molecules and on the endothelial monocyte adhesion. Differentiated Caco-2 monolayers were grown on the upper side of the inserts and placed in proximity to HMEC-1 grown on the bottom of the wells. GPE (5 and 10 μg/mL) was added on the apical compartment for 2 h, after which LPS (10 μg/mL) and TNF-α (10 ng/mL) were applied on the apical compartment and TNF-α (10 ng/mL) on the basolateral compartment for 16 h (**B**) or 12 h. (**A**,**C**) Endothelial cell surface expression of VCAM-1 and ICAM-1 was assessed by cell surface EIA and expressed as percentage of unstimulated control. (**B**) VCAM-1 and ICAM-1 mRNA levels were analyzed by quantitative RT-PCR and expressed as fold over unstimulated control (mean ± SD). (**C**) HMEC-1 were co-cultured with labeled THP-1 cells for 1 h. The number of adherent THP-1 cells was monitored by fluorescence microscope or measured by the fluorescence plate reader. Each experiment was performed in triplicate. The white histograms correspond to the untreated control (CTR), while the colored histograms correspond to the different treated groups (black for LPS/TNF-α and pink for GPE+LPS/TNF-α). # *p* < 0.01 versus control; * *p* < 0.05 versus LPS/TNF-α alone.

**Table 1 nutrients-14-01175-t001:** Oligonucleotides used for quantitative real-time PCR analysis.

Gene Name	Accession Number	Forward Primer	Reverse Primer	Size (bp)
IL1B	NM_000576.2	5′-CTGTCCTGCGTGTTGAAAGA-3′	5′-AGTTATATCCTGGCCGCCTT-3′	228
IL6	NM_000600.3	5′-AGGAGACTTGCCTGGTGAAA-3′	5′-CAGGGGTGGTTATTGCATCT-3′	180
TNF	NM_000594.2	5′-CCTGTGAGGAGGACGAACAT-3′	5′-AGGCCCCAGTTTGAATTCTT-3′	240
CXCL10	NM_001565.2	5′-CAAGGATGGACCACACAGAG-3′	5′-GCAGGGTCAGAACATCCACT-3′	248
CCL2/MCP1	NM_002982.3	5′-CCCCAGTCACCTGCTGTTAT-3′	5′-TCCTGAACCCACTTCTGCTT-3′	166
CSF1/MCSF	NM_000757.4	5′-TGGACGCACAGAACAGTCTC-3′	5′-CCTCCAGGGCTCACAATAAA-3′	235
PTGS2/COX-2	NM_000963.2	5′-TGCTGTGGAGCTGTATCCTG-3′	5′-GAAACCCACTTCTCCACCA-3′	176
VCAM1	NM_00107B.3	5′-CATGGAATTCGAACCCAAAC-3′	5′-CCTGGCTCAAGCATGTCATA-3′	140
ICAM1	NM_000201.2	5′-AGACATAGCCCCACCATGAG-3′	5′-CAAGGGTTGGGGTCAGTAGA-3′	190
MMP9	NM_004994.2	5′-AAAGCCTATTTCTGCCAGGAC-3′	5′-GTGGGGATTTACATGGCACT-3′	157
MMP2	NM_004530.4	5′-CACTTTCCTGGGCAACAAAT-3′	5′-TGATGTCATCCTGGGACAGA-3′	257
TIMP1	NM_003254.2	5′-TGACATCCGGTTCGTCTACA-3′	5′-CTGCAGTTTTCCAGCAATGA-3′	103
TIMP2	NM_003255.4	5′-CCAAGCAGGAGTTTCTCGAC-3′	5′-TTTCCAGGAAGGGATGTCAG-3′	121
GAPDH	NM_002046.3	5′-ATCACTGCCACCCAGAAGAC-3′	5′-TTCTAGACGGCAGGTCAGGT-3′	210
18 ribosomal RNA	NR_003286.2	5′-AAACGGCTACCACATCCAAG-3′	5′-CCTCCAATGGATCCTCGTTA-3′	155

**Table 2 nutrients-14-01175-t002:** Antioxidant activity and total phenols of GPE.

TEAC	ORAC	TP
µmolTE/g	µmolTE/g	mgGAE/g
794.98 ± 39.75	801.99 ± 40.09	61.40 ± 1.03

Data are expressed as mean value ± standard deviation and are representative of three independent experiments. TEAC: Trolox equivalent antioxidant capacity; ORAC: Oxygen radical absorbance capacity; TP: Total phenols.

**Table 3 nutrients-14-01175-t003:** GPE effect on the gene expression of inflammatory mediators in Caco-2 cells under inflammatory conditions.

Gene Name	LPS/TNF-α	GPE + LPS/TNF-α
IL1B	9.3 ± 0.8	7.5 ± 0.7 *
TNF	4.3 ± 0.7	0.9 ± 0.6 **
CXCL10	14.6 ± 1.2	5.9 ± 0.8 **
CSF1/MCSF	4.0 ± 0.6	1.6 ± 0.7 **
PTGS2/COX-2	3.8 ± 0.9	1.7 ± 0.6 *
VCAM1	15.1 ± 2.3	2.3 ± 0.8 **
ICAM1	1.56 ± 0.3	1.0 ± 0.2 *

The expression levels of genes were analyzed by quantitative RT-PCR. Data are reported as fold over unstimulated control (mean ± SD) and are representative of three independent experiments. * *p* < 0.05, ** *p* < 0.01 versus LPS/TNF-α alone.

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
