# Peer review of "Grape Pomace Extract Attenuates Inflammatory Response in Intestinal Epithelial and Endothelial Cells: Potential Health-Promoting Properties in Bowel Inflammation"

_nutrients, 2022, doi:10.3390/nu14061175_

Round 1
Reviewer 1 Report
This article by Calabriso, Carluccio, and co-workers describes the study of anti-inflammatory activities of polyphenolic compounds obtained from grape pomace extracts. The authors focused this research on enterocyte-like cells, especially from intestinal epithelial. The obtained results showed that polyphenols from these extracts can prevent inflammatory effects. The authors also determined that the effects of polyphenols are centered on the suppression of the gene expression of several pro-inflammatory mediators.
This is a well-done work that contributes to the knowledge of the possible application of polyphenols from grape pomace to the development of new therapeutic tools against inflammatory processes.
Additionally, a few considerations have to be taken into account.
- Some typing mistakes have been detected along with the text. An extensive revision is necessary.
Author Response
This article by Calabriso, Carluccio, and co-workers describes the study of anti-inflammatory activities of polyphenolic compounds obtained from grape pomace extracts. The authors focused this research on enterocyte-like cells, especially from intestinal epithelial. The obtained results showed that polyphenols from these extracts can prevent inflammatory effects. The authors also determined that the effects of polyphenols are centered on the suppression of the gene expression of several pro-inflammatory mediators.
This is a well-done work that contributes to the knowledge of the possible application of polyphenols from grape pomace to the development of new therapeutic tools against inflammatory processes.
Additionally, a few considerations have to be taken into account.
- Some typing mistakes have been detected along with the text. An extensive revision is necessary.
Response: We are honored for the positive comments. We thank the reviewer for the suggestions that allowed us to improve the manuscript. As suggested, the typing mistakes have been corrected and a wide revision of the text has been carried out.
Reviewer 2 Report
The manuscript entitled “Grape pomace extract attenuates overwhelming inflammatory response in intestinal epithelial and endothelial cells: potential health promoting properties in inflammatory bowel disease” observed anti-inflammatory effects of grape pomace in Caco2/HMEC1 cells as a potential absorption model. This manuscript contained the straightforward conclusion; however, this manuscript is not mechanistic. At least, authors should address the potential anti-inflammatory effects of grape pomace by which potential isomers with potential working mechanism. Therefore, discussion should be updated as a supportive structure to support the main idea of the manuscript.
- Title is redundant. Without in vivo data, it is hard to conclude that grape pomace attenuates inflammatory bowel disease. Recommend to discard “overwhelming” and “disease” from the title at least.
- Authors should address the potential anti-inflammatory effects of specific isomers in grape pomace listed in Figure 2 in the discussion part in detail. Authors may summarize the anti-inflammatory effects of individual isomers in the discussion with absorption rate.
- All controls do not have an error bars. Please add error bar for all controls.
- Please describe potential working mechanism in the discussion (i.e. suppression of NFkB activation, suppression of phosphorylation in MAPK-NFkB axis).
Author Response
The manuscript entitled “Grape pomace extract attenuates overwhelming inflammatory response in intestinal epithelial and endothelial cells: potential health promoting properties in inflammatory bowel disease” observed anti-inflammatory effects of grape pomace in Caco2/HMEC1 cells as a potential absorption model. This manuscript contained the straightforward conclusion; however, this manuscript is not mechanistic. At least, authors should address the potential anti-inflammatory effects of grape pomace by which potential isomers with potential working mechanism. Therefore, discussion should be updated as a supportive structure to support the main idea of the manuscript.
Response: We thank the reviewer for the suggestions that allowed us to improve the manuscript. We have carefully analyzed his concerns and considerations and answered to them.
1. Title is redundant. Without in vivo data, it is hard to conclude that grape pomace attenuates inflammatory bowel disease. Recommend to discard “overwhelming” and “disease” from the title at least.
Response: In accordance with the reviewer's suggestion, we rephrased the title as follows: “Grape pomace extract attenuates inflammatory response in intestinal epithelial and endothelial cells: potential health promoting properties in bowel inflammation”
2. Authors should address the potential anti-inflammatory effects of specific isomers in grape pomace listed in Figure 2 in the discussion part in detail. Authors may summarize the anti-inflammatory effects of individual isomers in the discussion with absorption rate.
Response: In accordance with the reviewer's comments, in the discussion of the revised manuscript we have reported data on the rate of absorption and the anti-inflammatory effects of the individual polyphenols present in the grape pomace extract (lines 616-636)
3. All controls do not have an error bars. Please add error bar for all controls.
Response: As suggested, in the figures of the revised manuscript we have added the error bar for all the controls.
4. Please describe potential working mechanism in the discussion (i.e. suppression of NFkB activation, suppression of phosphorylation in MAPK-NFkB axis).
Response: We appreciate the reviewer’s comment. As suggested, in the discussion of the revised manuscript we have reported data on the mechanism of action and the signalling pathways of the individual polyphenols present in the grape pomace extract (lines 568-593).
Round 2
Reviewer 2 Report
The authors updated their manuscript in a sound way.